# Resource Allocation in C-V2X Mode 3 Based on the Exchanged Preference Profiles

**Dizhe Yuan \*** , **Denghua Hu and Xihong Chen**

Air and Missile Defense College, Air Force Engineering University, Xi'an 710051, China
\* Correspondence: yuandizhe2018@163.com

**Abstract:** In this paper, we investigate the resource block (RB) allocation problem in cellular vehicle-to-everything (C-V2X) networks mode 3, where the cellular networks schedule the RBs for direct vehicular communications. First, we establish the communication model and introduce the effective capacity and queuing theory to describe the reliability of vehicle-to-vehicle (V2V) links. Then, we introduce the $\alpha$-fair function and formulate the joint power control and RB allocation problem considering the allocation fairness and the different quality-of-service (QoS) requirements for vehicle-to-infrastructure (V2I) and V2V links. Our objective is to maximize the sum capacity of all V2I links with the $\alpha$-fair function while guaranteeing the allocation fairness among V2I links and the transmission reliability for each V2V pair. To achieve this objective, we propose a novel matching game theory algorithm based on the exchanged preference profiles between the two participant sets, i.e., V2V and V2I links. Simulation results show that our proposed algorithm is adaptive to the dynamic vehicular network and achieves better efficiency and fairness trade-offs, outperforming the classic allocation method.

**Keywords:** C-V2X; mode 3; resource allocation fairness; matching game theory; preference profiles

## 1. Introduction

As a key enabler for intelligent transportation systems (ITSs), vehicular networks have attracted considerable attention in recent years due to the potential to improve road safety and traffic efficiency and support infotainment requirements. There are two prominent communication technologies for vehicular networks: dedicated short-range communications (DSRC) and cellular vehicle-to-everything (C-V2X) [1,2]. DSRC is supported by the IEEE 802.11p standard and faces many challenges, such as limited mobility support and unacceptable latency under a high vehicle density. As an alternative to DSRC, C-V2X possesses various advantages, including strong scalability, a high data rate, and a quality of service (QoS) guarantee [3,4]. In release 14 [5], the 3GPP developed two operational modes for direct vehicular communications in C-V2X: mode 3 and mode 4. In mode 4, vehicles do not require the cellular coverage and autonomously select and reserve the subchannel resources for information transmissions using a distributed scheduling scheme, while in mode 3, the orthogonal subchannel resources are scheduled by the cellular networks. Compared to mode 4, the scheduling scheme of subchannel resources is more efficient in mode 3 due to the comprehensive knowledge collected by the cellular network from all vehicles in its coverage.

The high mobility in vehicular networks causes channel conditions to change rapidly over time, impairing communication efficiency [6,7]. Thus, the resource allocation method plays a vital role in vehicular networks and needs to cater for the varying channel conditions to improve and maintain communication efficiency.

### 1.1. Related Work

Most recent research in the design of resource allocation method in C-V2X mainly focuses on the underlay spectrum approach, in which the cellular user equipment (CUE)

and the vehicular user equipment (VUE) utilize the same spectrum band at the same time, achieving a higher spectrum efficiency. In [8], the different QoS requirements for vehicle-to-infrastructure (V2I) and vehicle-to-vehicle (V2V) links were analyzed, and the ergodic capacity of the V2I was calculated only through the large-scale fading information. Then, Ref. [8] studied the optimal power allocation for each reused CUE-VUE pair and used the Hungarian algorithm to perform the spectrum resource assignment. Ref. [9] improved the vehicle platooning communication model by considering both the V2V communication and controlling factors and designed the corresponding cooperative awareness messages (CAMs) dissemination mechanism based on the bipartite graph matching. Ref. [10] formulated the resource allocation problem in vehicular networks as a latency-minimized problem, the objective of which was to maximize the weighted sum of the latency reductions. The expected latency and packet delivery ratio performances were analyzed in [10]. Ref. [11] utilized the queuing theory and effective capacity to formulate the latency violation probability (LVP) for the V2V link and maximize the V2I links' sum ergodic capacity under the constraint of the LVP of the V2V link. Ref. [12] clustered the vehicles into coalitions to acquire the benefits of spatial reuse and studied the power allocation for vehicular uplink networks. Ref. [13] analyzed the impact of vehicles' transmitting power on the packet delivery ratio performance of mode 4 and proposed an adaptive-transmit power control algorithm to reduce interference among neighboring vehicles. Ref. [4] classified the spectrum sharing model in mode 3 into two types, i.e., overlay and underlay approaches, and formulated the resource allocation problem as the optimization problem of minimizing the number of unallowed vehicular links. To transform the nonconvex optimization problem into a convex problem, Ref. [4] applied the McCormick envelopes method to linearize the corresponding variables. In [14], the closed optimal power allocation for a single pair of a vehicle and a cellular user was derived, and the three-partite hypergraph was utilized to allocate resource blocks (RBs) for multiple CUEs and VUEs. Ref. [15] proposed a location-based maximum reuse distance (MRD) scheduling method based on the allocation scheme in [16], which considered the locations of vehicles applying for RBs reuse. Ref. [17] analyzed the four types of conditions that avoided the resource allocation conflicts in mode 3, which were the differentiated QoS per vehicle, precluding the intracluster subframe conflicts, guaranteeing the minimal time dispersion, and preventing the concurrent signals received by the vehicles lying at the intersection, respectively. The corresponding mathematical framework of the subchannel allocation based on the four conditions was formulated in [17]. Considering the stringent QoS reliability requirements of the vehicular applications, Ref. [18] proposed a priority- and guarantee-based resource allocation method, which first guaranteed the minimum RBs for safety applications and then prioritized the emergency messages to allocate the remaining radio resources properly. Ref. [19] formulated the resource allocation problem as a Stackelberg game and defined the corresponding game equilibrium concept to evaluate the allocation results. Based on the price–penalty mechanism, the selfish behaviors of vehicle pairs could be limited. Ref. [20] studied resource allocation fairness and the various QoS demands among mobile device-to-device (D2D) users and established an interference graph to improve energy efficiency and guarantee the users' requirements, which was based on the graph coloring theory. Ref. [21] proposed two new spectrum-repartitioning and frequency-reuse techniques in roadside units for vehicular communications, namely full frequency reuse and partial frequency reuse, to reduce the number of packet collisions in broadcast transmission. Ref. [22] utilized the fuzzy interference system theory and designed the corresponding matching statement table to dynamically determine the resource keep probability in mode 4 to improve the packet delivery ratio performance. In [23–25], deep reinforcement learning was introduced into the resource allocation in mode 3 to improve the latency performance, and the corresponding action space, the state space, and the reward function were designed for different communication scenarios.

As pointed out above, most previous studies in the spectrum resource allocation method design generally adopt the Hungarian algorithm and the machine learning method.

The Hungarian algorithm stipulates that the number of VUEs and CUEs must be equal, making it hard in practice. In the machine learning method for mode 3, the management and optimization of radio resources generally involve multiple vehicles with various QoS requirements, leading to an explosive growth in spatial dimensionality and significant increases in computation complexity. Fortunately, those deficiencies could be tackled with game theory. Matching game theory is a powerful method in game theory that can well describe the interactions and mutually beneficial relations between different participant sets. The preference profiles defined in matching game theory can handle the multiple QoS requirements of the vehicles, and there are no restrictions on the number of agents from two different participant sets. Moreover, the corresponding matching algorithm possesses a low computational complexity and thus can respond quickly to the highly dynamic vehicular network structure. Although the matching game theory has been applied in [26–28], these studies do not consider the efficiency and fairness trade-offs in the resource allocation in mode 3.

### 1.2. Contribution and Organization

In this paper, our objective is to maximize the sum capacity of all CUEs while guaranteeing allocation fairness among CUEs and the transmission reliability for each VUE pair. Note that we are using the terms CUEs and V2I links, VUEs and V2V links interchangeably in this paper. Our main contributions include the following:

1. We utilize the effective capacity theory and LVP to measure the reliability of the V2V link and formulate the joint power control and RBs resource allocation problem with the objective of maximizing the sum data rate of CUEs with the $\alpha$-fair function. The formulated problem is a mixed-integer nonlinear programming (MINLP) problem of NP-hard complexity.

2. To solve the formulated problem, we propose a novel matching algorithm based on the exchanged preference profiles to obtain the optimal spectrum allocation results using the matching game theory. We also analyze the optimality and convergence of our proposed algorithm.

3. Extensive simulations are conducted to evaluate our proposed algorithm's fairness, achievable throughput, and outage ratio performance under different vehicular networks. The simulation results show that our proposed algorithm is highly affected by the parameter $\alpha$ in the $\alpha$-fair function and can obtain better efficiency and fairness trade-offs compared to the classical allocation method in [8].

The rest of this paper is organized as follows. Section 2 establishes the system model. Section 3 introduces the $\alpha$-fair function and formulates the joint power control and spectrum resource allocation problem. The resource allocation algorithm is presented in Section 4. Section 5 discusses and analyzes the simulation results, and Section 6 concludes the paper.

## 2. System Model

In this section, we first establish the system communication mode and then introduce the V2V links' effective capacity conception to evaluate the corresponding reliability and latency performance.

### 2.1. Communication Model

We consider a single-cell C-V2X network with one BS denoted by $C$ consisting of multiple CUEs and VUEs. Let $\mathcal{M} = \{1, 2, \ldots M\}$ be the set of $M$ CUEs and $\mathcal{K} = \{1, 2, \ldots K\}$ be the set of $K$ VUEs. The basic resource allocation unit is defined as the RBs. $\mathcal{N} = \{1, 2, \ldots N\}$ denotes the set of RBs. In our RBs scheduling scheme, due to the high capacity requirement for the V2I links, a CUE can occupy multiple RBs. Moreover, to avoid the interference between different CUEs, each RB can be assigned to only one CUE. As the uplink spectrum is less intensive and the VUEs usually generate tolerable interference with the BS, we stipulate that VUEs can reuse the RBs assigned to CUEs.

As shown in Figure 1, the channel power gain from CUE $m$ to the BS on RB $n$, denoted by $h_{m,n}^C$, is formulated as $h_{m,n}^C = \alpha_{m,n}^C g_{m,n}^C$, where $\alpha_{m,n}^C$ and $g_{m,n}^C$ account for the corresponding large-scale and small-scale fading, respectively. The channel power gain $h_{k,n}^V$ from VUE $k$ to its V2V pair on RB $n$, the interfering channel $h_{m,k,n}^V$ from CUE $m$ to VUE $k$ on RB $n$, and interfering channel $h_{k,m,n}^C$ from the VUE $k$ to CUE $m$ on RB $n$ are defined similarly. Due to the high mobility of vehicles, we assume that the BS can acquire the knowledge of the large-scale fading periodically and only be aware of the statistical characterization of the small-scale components rather than their realizations.

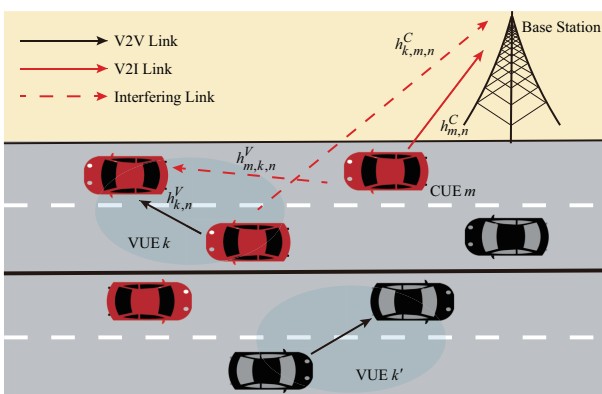

**Figure 1.** C-V2X mode 3 communication model.

The signal-to-interference-plus-noise ratio (SINR) of the CUE $m$ on RB $n$ is expressed as

$$\gamma_{m,n}^C = \frac{x_{m,n} P_{m,n}^C \alpha_{m,n}^C g_{m,n}^C}{\sigma^2 + \sum\limits_{k \in \mathcal{K}} x_{k,n} P_k^V \alpha_{k,m,n}^C g_{k,m,n}^C} \tag{1}$$

where the binary variable $x_{m,n}$ represents the allocation indicator of RB $n$. If $x_{m,n}$ is equal to 1, CUE $m$ is allocated RB $n$, and 0 otherwise. The definition of $x_{k,n}$ is similar to that of $x_{m,n}$. $P_{m,n}^C$ and $P_k^V$ denote the transmit powers of CUE $m$ on RB $n$ and VUE $k$, respectively. $\sigma^2$ is the power of the received additive white Gaussian noise (AWGN).

Similarly, the SINR of the VUE $k$ on RB $n$ can be defined as

$$\gamma_{k,n}^V = \frac{x_{k,n} P_k^V \alpha_{k,n}^V g_{k,n}^V}{\sigma^2 + \sum\limits_{k \in \mathcal{K}} x_{m,n} P_{m,n}^C \alpha_{m,k,n}^V g_{m,k,n}^V} \tag{2}$$

Since different VUEs may be in close distances to each other, we stipulate that the VUEs can only be assigned one RB and cannot occupy the same RBs.

Due to the high vehicles mobility, the small-scale fading components of the channel power gain vary on a highly fast scale. Thus, the achievable data rate of the CUE $m$ on RB $n$ can be expressed as the ergodic capacity, which is

$$\mathbb{E}\left\{ R_{m,n}^C \right\} = \mathbb{E}\left\{ W \log(1 + \gamma_{m,n}^C) \right\} \tag{3}$$

where $W$ is the bandwidth of the single RB. Let $\mathbb{E}\left\{ R_m^C \right\}$ denote the total ergodic capacity of the CUE $m$, which is $\mathbb{E}\left\{ R_m^C \right\} = \sum\limits_{n \in \mathcal{N}} \mathbb{E}\left\{ R_{m,n}^C \right\}$.

## 2.2. V2V Links' Effective Capacity

As can be observed in Formula (2), the SINR of the V2V link is dynamic in different slots, leading to the instability of the network serviceability. Thus, in this paper, we assume that the traffic generated by the VUE enters an infinite-size first-in-first-out (FIFO) buffer before it is transmitted through the V2V channel, which is shown in Figure 2. Since the vehicles periodically broadcast the CAMs to inform the neighbors of their presence, the

generated data traffic rate of the VUE can be regarded as a constant $\lambda_k$ and the source traffic of VUE $k$ during $[0,t]$ is $A_k(t) = \lambda_k t$. We define the actual service traffic as $S_k(t)$, and the achievable service traffic is $\hat{S}_k(t)$. It is obvious that $S_k(t)$ is less than or equal to $A_k(t)$ for any $t > 0$ and $S_k(t)$ is bounded by the minimum value of $A_k(t)$ and $\hat{S}_k(t)$, which can be expressed as

$$S_k(t) = \min\{\hat{S}_k(t), A_k(t)\} \tag{4}$$

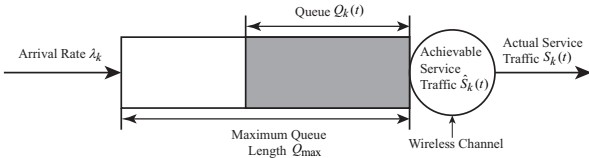

**Figure 2.** Queue theory model.

Figure 3 illustrates the LVP of the V2V link. As shown in Figure 3, $D_{max}$ represents the maximum tolerable latency of packet transmission, and $Q_{max}$ is the maximum traffic queue length in the buffer. The minimum traffic curve, which should be served to guarantee a transmission latency of no more than $D_{\max}$, is expressed as $\psi(t) = \lambda_k(t - D_{\max})$. The latency violation events happen when the actual service curve $S_k(t)$ is lower than the curve $\psi(t)$. Thus, the LVP of the V2V link $k$ can be expressed as

$$\Pr\{D_k \geq D_{\max}\} = \frac{\int_{t \in \mathcal{V}} dS_k(t)}{S_k(lT)} \tag{5}$$

where $\mathcal{V}$ denotes the part where $S_k(t) \leq \psi(t)$ and $D_k$ is the latency experienced by the VUE $k$. $T$ is the length of one time slot, and $l \in \{0, 1, 2, \dots\}$.

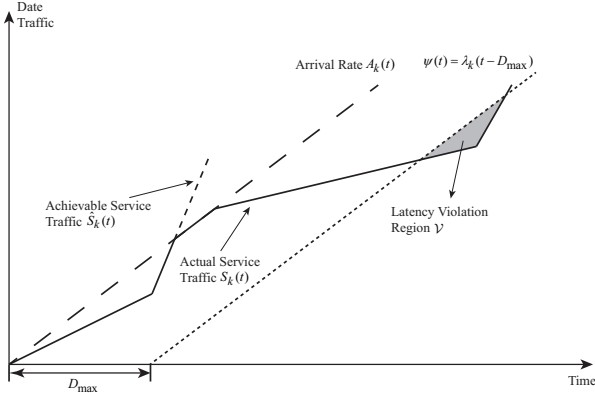

**Figure 3.** Traffic and service model.

Formula (5) reflects that the LVP of a V2V link is closely related to the maximum tolerable latency $D_{max}$ and the channel's actual cumulative service traffic $S_k(t)$. Thus, to characterize the relations between $D_{max}$ and $S_k(t)$, according to [11], we introduce the effective capacity conception $EC_k(\theta_k^V)$, which provides a measure for the maximum constant source traffic rate supported by a given service rate under the constraint of latency requirement factor $\theta_k^V$. The effective capacity $EC_k(\theta_k^V)$ of VUE $k$ can be expressed as

$$EC_k(\theta_k^V) = -\frac{1}{\theta_k^V T} \ln E\left\{e^{-\theta_k^V TW\log_2(1+\gamma_{k,n}^V)}\right\} \tag{6}$$

As proved in [11], the effective capacity $EC_k(\theta_k^V)$ is monotonically decreasing with $\theta_k^V \in [0, \infty)$. The effective capacity reaches its maximum value when $\theta_k^V = 0$, which is

$$EC_k(\theta_k^V = 0) = \mathbb{E}\left\{W\log(1 + \gamma_{k,n}^V)\right\} \tag{7}$$

Moreover, the effective capacity reaches its minimum value when $\theta_k^V = \infty$, i.e., $EC_k(\theta_k^V = \infty) = 0$.

Particularly, when the source traffic rate $\lambda_k$ is equal to $EC_k(\theta_k^V)$, the LVP of the VUE $k$ can be expressed as

$$\Pr\{D_k \geq D_{\max}\} = p(\lambda_k)e^{-\theta_k^V \lambda_k D_{\max}} \tag{8}$$

where $p(\lambda_k)$ denotes the buffer nonempty probability in the steady queue state, which is

$$p(\lambda_k) = \frac{\lambda_k}{\mathbb{E}\left\{W\log_2(1 + \gamma_k^V)\right\}} \tag{9}$$

## 3. Problem Formulation

In this section, we formulate the joint power control and RBs assignment problem to maximize the data rate of CUEs with the $\alpha$-fair utility function while guaranteeing the minimum reliability for each VUE.

### 3.1. $\alpha$-Fair Utility Function

To improve the resource allocation fairness among CUEs, we introduce the $\alpha$-fair utility function [29], which is defined as

$$U_\alpha(x) = \begin{cases} \log(x), & \alpha = 1 \\ x^{1-\alpha}/(1-\alpha), & 0 \leq \alpha < 1 \end{cases} \tag{10}$$

where the parameter $\alpha \in [0,1]$ and characterizes the trade-off between the allocation efficiency and fairness.

As observed in (10), the $\alpha$-fair function is a nondecreasing concave-down function, the slope of which decreases as $x$ increases. Thus, for the $\alpha$-fair function, the increases in low rates are more favored than the larger rates. For example, an RB is scheduled to be assigned to $CUE_1$ or $CUE_2$, the current transmission capacities of which are $x_1$ and $x_2$, respectively. Since the derivatives $U_\alpha'(x_1)$ and $U_\alpha'(x_2)$ of the utility function satisfy $U_\alpha'(x_1) > U_\alpha'(x_2)$ for $x_1 < x_2$, the RB allocated to $CUE_1$ will obtain more system utility values, achieving allocation fairness between $CUE_1$ and $CUE_2$. Moreover, the larger the value of $\alpha$, the better the fairness of the resource allocation results.

### 3.2. Problem Definition

The special characteristics of the resource allocation problem in the C-V2X mode 3 network include two main aspects: the high vehicle mobility and the different requirements for different types of links. Considering the rapid variations in channels' conditions caused by the high vehicle mobility, our proposed resource allocation scheme depends solely on the slowly varying large-scale channel parameters and the distributions of the small-scale channel parameters, thus significantly reducing the signaling overheads and making the scheme more practical. On the other hand, the V2I connections desire large transmission capacity and allocation fairness among CUEs, while V2V connections place greater emphasis on link reliability and low communication latency. Thus, the resource allocation problem can be formulated as

$$\max \sum_{m \in \mathcal{M}} \sum_{n \in \mathcal{N}} x_{m,n} U_\alpha \left( \mathbb{E} \left\{ R_{m,n}^C \right\} \right)$$

$s.t.$

C1: $\quad P\{D_k > D_{\max}\} \geq p_0, \ \forall k \in \mathcal{K}$

C2: $\quad \lambda_k = -\dfrac{1}{\theta_k^V T} \ln \mathbb{E} \left\{ e^{-\theta_k^V TW \log_2(1+\gamma_{k,n}^V)} \right\}, \ \forall k \in \mathcal{K}$

C3: $\quad \sum_{n \in \mathcal{N}} x_{m,n} \mathbb{E} \left\{ R_{m,n}^C \right\} \geq R_0, \ \forall m \in \mathcal{M}$

C4: $\quad 0 \leq \sum_{n \in \mathcal{N}} x_{k,n} \leq 1, \ \forall k \in \mathcal{K}$ $\qquad$ (11)

C5: $\quad 0 \leq \sum_{n \in \mathcal{N}} x_{m,n} \leq n_q, \ \forall m \in \mathcal{M}$

C6: $\quad 0 \leq P_k^V \leq P_{\max}^V, \ \forall k \in \mathcal{K}$

C7: $\quad 0 \leq \sum_{n \in \mathcal{N}} x_{m,n} P_{m,n}^C \leq P_{\max}^C, \ \forall m \in \mathcal{M}$

C8: $\quad 0 \leq P_{m,n}^C \leq P_{n,\max}^C, \ \forall m \in \mathcal{M}, \forall n \in \mathcal{N}$

where $p_0$ is the LVP threshold of the VUEs, and $R_0$ denotes the minimum ergodic capacity requirement of the CUE. Constraints C1 and C3 represent the reliability requirements and minimum capacity for each VUE and CUE, respectively. Constraint C2 stipulates that the channel's effective capacity is equal to the source data rate. Constraint C4 presents that each VUE can be assigned at most one RB. Constraint C5 models our assumption that each CUE is assigned at most $n_q$ RBs. $n_q$ denotes the quota of a CUE, which can be calculated by $\lfloor P_{\max}^C / P_{n,\max}^C \rfloor$. Constraints C6 and C8 represent the VUE's and CUE's maximum power limits, and constraint C7 stipulates the maximum transmission power on an RB of the CUE, which is denoted by $P_{n,\max}^C$.

## 4. Resource Allocation via Matching

As can be observed in (3.2), the formulated problem is a nonconvex MINLP problem, which is difficult to solve in practical settings with large sets of vehicles and RBs. Since the interference only exists within each reuse pair consisting of a CUE and a VUE, problem (3.2) can be decomposed into two subproblems, which are power control and spectrum allocation. The power control subproblem has been addressed in [4,8]. Hence, in this paper, we only consider the spectrum resource allocation for problem (3.2). We define $\mathbb{E}\{R_{m,k}^{C\,*}\}$ as the optimal ergodic capacity solutions of the power control subproblem. $R_{m,k}^{C\,*}$ is the transmission capacity when CUE $m$ reuses RB $n$ with VUE $k$ under the optimal power control $\{P_{m,n}^{C\,*}, P_k^{V\,*}\}$, which is

$$R_{m,k}^{C\,*} = W \log \left( 1 + \frac{P_{m,n}^{C\,*} \alpha_{m,n}^C g_{m,n}^C}{\sigma^2 + P_k^{V\,*} \alpha_{k,m,n}^C g_{k,m,n}^C} \right) \qquad (12)$$

Then, problem (3.2) can be transformed into:

$$\max \sum_{m\in\mathcal{M}}\sum_{k\in\mathcal{K}} x_{m,k} U_\alpha\left(\mathbb{E}\left\{R_{m,k}^{C}{}^*\right\}\right)$$

$$s.t.$$

$$\text{C1:} \quad 0 \le \sum_{m\in\mathcal{M}} x_{m,k} \le 1, \ \forall k \in \mathcal{K} \tag{13}$$

$$\text{C2:} \quad 0 \le \sum_{k\in\mathcal{K}} x_{m,k} \le n_q, \ \forall m \in \mathcal{M}$$

$$\text{C3:} \quad x_{m,k} \in \{0,1\}, \ \forall m \in \mathcal{M}, \forall k \in \mathcal{K}$$

where constraint C1 represents that one VUE is allowed to access the RBs of a single CUE, and constraint C2 represents that the spectrum of one CUE can be shared with at most $n_q$ VUEs.

*4.1. Matching-Game Formulation*

As can be observed from problem (13), the two participating agents in the matching process are the CUE set $\mathcal{M}$ and the VUE set $\mathcal{K}$, respectively.

Since problem (13) is also an MINLP problem, to address this challenge, we formulate problem (13) as a many-to-one matching game with two-sided preferences and propose a novel matching algorithm based on the exchanged preference profiles. The many-to-one matching means that each agent from the CUE set can be matched to more than one member from the VUE set, while members from the VUE set can only be matched to at most one CUE. The two-sided preferences mean that each agent of the VUE or CUE set ranks the members in the opposite set in the order of its preference. The definition of the corresponding matching $\mu$ is as follows:

**Definition 1.** *A matching game $\mu$ is defined by two sets of players $\mathcal{M} \cup \mathcal{K}$, which satisfies for $\forall k \in \mathcal{K}, m \in \mathcal{M}$:*

1. *If $k \in \mu(m)$, then $\mu(k) = m$.*
2. *If $\mu(k) = m$, then $\mu(m) \in \mathcal{M}_{k,m}$.*
3. *$|\mu(k)| \le 1, 0 \le |\mu(m)| \le n_q$.*

*where $\mathcal{M}_{k,m}$ denotes the set of vehicles who prefers the CUE $m$, and $|\cdot|$ is the cardinality of a set.*

*4.2. Exchanged Preference Profiles*

In the C-V2X network, each VUE establishes the preference profile to rank the different CUEs, and each CUE possesses certain preferences for the VUEs which are acceptable. In our proposed game, we exchange the preference profiles of the two sides, which means that a VUE chooses the match according to the benefits of CUEs, and the preference profiles of CUEs are established based on the benefits of VUEs.

The preference profile for the VUE $k$ is defined as follows:

$$U_k(m) = U_\alpha\left(\mathbb{E}\left\{\sum_{k\in\mathcal{K}} x_{m,k}(t+1)R_{m,k}^{C}{}^*(t+1)\right\}\right) - U_\alpha\left(\mathbb{E}\left\{\sum_{k\in\mathcal{K}} x_{m,k}(t)R_{m,k}^{C}{}^*(t)\right\}\right) \tag{14}$$

where the time slot $t$ indicates a matching decision made by VUE $k$.

Since the LVP is monotonically decreasing with respect to VUE $k$'s effective capacity, the preference profile of the CUE $m$ is based on the achievable data rate of the VUE $k$ when they share the RB $n$.

$$U_m(k) = \mathbb{E}\left\{W\log\left(1 + \gamma_{k,m}^{V}{}^*\right)\right\} \tag{15}$$

where

$$\gamma_{k,m}^{V}{}^{*} = \frac{P_k^{V*}\alpha_{k,n}^{V}g_{k,n}^{V}}{\sigma^2 + P_{m,n}^{C}{}^{*}\alpha_{m,k,n}^{V}g_{m,k,n}^{V}} \tag{16}$$

The reason why we exchange the preference profiles of the two sides is as follows. Suppose the CUE adopts the preference profile defined as Equation (14). In that case, it chooses the reused vehicle creating the least interference, which has no differences from the traditional method targeting maximizing the sum data rate of CUEs. However, when the DUE adopts the preference profile (14), the reused CUE producing more $\alpha$-fair function increments is preferred. Considering the characteristics of the $\alpha$-fair function, the CUEs with a lower ergodic capacity gain dominance in the preference profiles of DUEs, leading to the assignment fairness. This illustrates an important and interesting fact that the system's fairness can only be achieved by putting one side in the other side's standpoint.

In addition, we define the preference relation $\succ_V$ of VUEs as

$$m\succ_V m' \Rightarrow U_k(m) > U_k(m') \tag{17}$$

Similarly, the preference relation $\succ_C$ of CUEs is defined as

$$k\succ_C k' \Rightarrow U_m(k) > U_m(k') \tag{18}$$

To address problem (13), we propose a resource allocation algorithm based on the exchanged preference profiles. Before presenting the algorithm, we define the stable matching allocation to evaluate the matching results.

**Definition 2.** *A matching $\mu$ is said to be stable if there exists no blocking pair $(m,k)$ satisfying the following conditions, $\forall k \in \mathcal{K}, m \in \mathcal{M}$:*

1. *$k$ is unassigned or $m\succ_V\mu(k)$;*
2. *$m$ is unassigned or $k\succ_C\mu(m)$.*

The stability ensures that no matched pair would benefit by deviating from their current matching decisions. As shown in constraint C2 of problem (13), we stipulate that each CUE possesses a specific quota for the assigned VUEs. Thus, to eliminate the influence of externalities, we propose Algorithm 1 to stabilize the matching result. Algorithm 1 is divided into the initialization phase and the matching phase. In the initialization phase, the BS establishes the preference profiles for each VUE and CUE using the channel state and local information.

In the matching phase, each VUE proposes to its most-preferred CUE according to the preference profile, and the most-preferred CUE is removed from the VUE's preference list. If CUE $m$ does not have a sufficient quota, CUE $m$ ranks the proposed VUE $k$ and its current matching VUEs. If VUE $k$ is ranked higher than the current match, the least-preferred current match is deleted, and VUE $k$ is accepted. Otherwise, it is rejected. The deleted and rejected VUEs are removed from CUE $m$'s preference list. If CUE $m$ has a sufficient quota, VUE $k$ is accepted directly. It is noted that if multiple VUEs propose to the same CUE simultaneously, the CUE reserves its most-preferred one and sends a signal to the others. Those VUEs receiving the signal add the CUE to their preference profiles again to eliminate the externalities. The algorithm does not terminate until the two consecutive matching results remain unchanged. The stability can be proved by Theorem 1.

**Theorem 1.** *Algorithm 1 converges to a stable allocation.*

---

**Algorithm 1** Spectrum Resource Allocation Algorithm

---

1: **Initialization:** Establish the preference profiles of VUEs $\mathcal{P}_k$ and CUEs $\mathcal{P}_m$. Set the rejected set $\mathcal{L}_m = \varnothing$ and the unassigned set $\mathcal{A} = \mathcal{K}$. Set the initial time slot $t = 1$ and the candidate set $\mathcal{C}_m = \varnothing$, which denotes the VUEs propose to CUE $m$ simultaneously.

2: **while** $\mu^{(t)} \neq \mu^{(t-1)}$ **do**

3:    $t \leftarrow t + 1$.

4:    **for** $\forall k \in \mathcal{A}$ **do**

5:       VUE $k$ proposes to its most-preferred CUE $m$ according to $\mathcal{P}_k^{(t)}$, and removes CUE $m$ from $\mathcal{P}_k^{(t)}$.

6:       **if** $\left| \mathcal{C}_m^{(t)} \right| > 1$ **then**

7:          $\mu^{(t)}(m) = k^*, k^* \succ_C k', \forall k' \in \mathcal{C}_m^{(t)} \backslash k^*$.

8:          $\mathcal{P}_{k'}^{(t)} \leftarrow \mathcal{P}_{k'}^{(t)} \cup m$.

9:       **end if**

10:      **if** $\left| \mu^{(t)}(m) \right| = n_q$ **then**

11:         **if** $\exists k' \in \mu^{(t)}(m), k \succ_C k'$ **then**

12:            $\mu^{(t)}(m) \leftarrow \mu^{(t)}(m) \backslash k'$.

13:            $\mu^{(t)}(m) = \mu^{(t)}(m) \cup k$.

14:            $\mathcal{L}_m^{(t)} \leftarrow \mathcal{L}_m^{(t)} \cup k', \mathcal{A} \leftarrow \mathcal{A} \backslash k$.

15:         **else**

16:            $\mathcal{L}_m^{(t)} \leftarrow \mathcal{L}_m^{(t)} \cup k$.

17:         **end if**

18:       **else**

19:         $\mu^{(t)}(m) = \mu^{(t)}(m) \cup k$.

20:         $\mathcal{A} \leftarrow \mathcal{A} \backslash k$.

21:       **end if**

22:    **end for**

23:    Update the preference profile $\mathcal{P}_k^{(t)}, \forall k \in \mathcal{A}$.

24:    $\mathcal{P}_k^{(t)} \leftarrow \mathcal{P}_k^{(t)} \backslash m, k \in \mathcal{L}_m^{(t)}, \forall m \in \mathcal{M}$.

25: **end while**

---

**Proof.** The stability of Algorithm 1 can be proved by contradiction. Assume the final matching result is not stable, and there exists only one VUE $k$ and one CUE $m$ satisfying $m \succ_V \mu(k)$ and $k \succ_C k'$ simultaneously. $k'$ is the least-preferred VUE in the current matchings of CUE $m$, i.e., $k' \in \mu(m)$. According to Algorithm 1, we have two cases as follows:

1. $|\mu(m)| \leq n_q$
   Since VUE $k$ prefers CUE $m$ to $\mu(k)$, it proposes to CUE $m$. For case 1, CUE $m$ possesses sufficient quota, and thus the VUE $k$ turns to match the CUE $m$, leading to a stable matching result again.

2. $|\mu(m)| = n_q$
   For case 2, CUE $m$ does not have a sufficient quota. After VUE $k$ proposes to CUE $m$, CUE $m$ ranks the proposed VUE $k$ and its current matching VUEs. Since $k \succ_C k'$, CUE $m$ deletes the VUE $k'$ and accepts the VUE $k$.

The matching results of the two cases above contradict our assumption, which illustrates that our proposed algorithm converges to a stable matching. □

In Algorithm 1, the computational complexity can be divided into two parts: the complexity of building the preference profiles and the running time complexity. For each VUE, the computational complexity of building the preference profiles is $O(M \log M)$. Similarly, the corresponding complexity for each CUE is $O(K \log K)$. Thus, the total computational complexity of building both participant sets' preference profiles is $O(KM \log KM)$. Under the worst case, the VUE proposes to all CUEs and the running time complexity is linear with the size of input participant sets, which is $O(KM)$. Therefore, the com-

putational complexity of Algorithm 1 is $O(KM \log(KM) + KM)$, which is reasonable for a practical implementation.

## 5. Numerical Results and Discussion

### 5.1. Scenario Setup

In this section, the objective was to validate the resource allocation efficiency and fairness performance of our proposed algorithm compared to the minimum capacity maximization (MCM) algorithm in [8] and the random selection scheme. We used the Jain's fairness index to measure the fairness performance of different algorithms [30]. The MCM algorithm was a max-min scheduler that exploited the Hungarian method and bisection search, and the spectrum-sharing assignment of the MCM algorithm was what the Hungarian method yielded when the bisection search ended. The Hungarian method of the MCM algorithm was to find the minimum capacity among all CUEs, and the bisection search was to maximize the minimum capacity value. In the random selection scheme, the reused pairs between the CUEs and VUEs were selected randomly.

We considered a highway segment of 800 m with three lanes in each direction, and the road width was set to 3 m. The BS was located at the midpoint of the highway and 35 m away from the road. The vehicles were dropped on the highway segment according to the Poisson process, and a vehicle could only be a CUE or a VUE. The CUEs and VUEs were randomly selected among the generated vehicles, and the V2V links were formed between neighboring vehicles. The vehicle's mean velocity was 70 km/h, and the standard deviation of the velocity was 10 km/h. The simulation scenario setup is shown in Figure 4. We assumed that the vehicles' velocities remained constant during the communication process. The channel models for V2V and V2I links were consistent with those of [8].

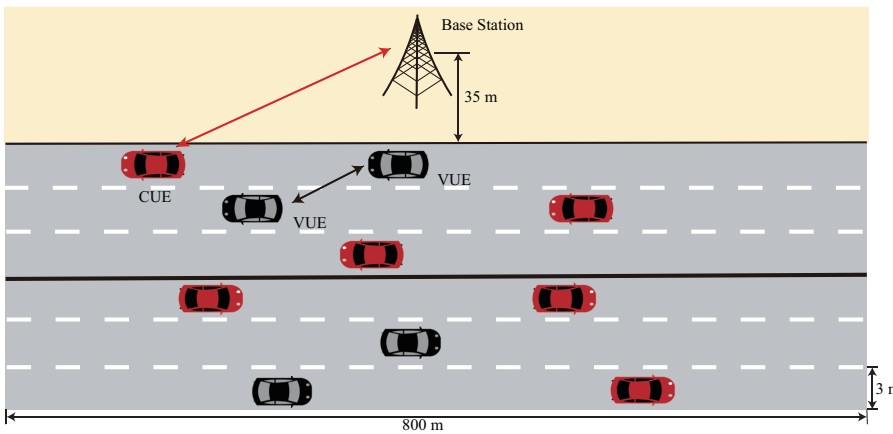

**Figure 4.** Simulation scenario setup.

The maximum tolerable latency $D_{max}$ was set to 0.001 s, the time slot length was 0.001 s, and the LVP threshold $p_0$ was 0.001. The bandwidth of each RB $W$ was 180 kHz, and the number of RBs was 100. The maximum total transmission powers $P_{max}^C$ and $P_{max}^V$ were set to 23 dBm, and the limited transmission power $P_{n,max}^C$ was equal to $P_{max}^C/3$. The power of AWGN was assumed to be $-114$ dBm, and the generated traffic data rate $\lambda_k$ was 2 Mbps. The minimum ergodic capacity requirement of the CUE $R_0$ was 0.5 bps/Hz.

### 5.2. Network Capacity Performance

In this subsection, we evaluated the achievable network capacity of our proposed algorithm by a comparison with the MCM algorithm and the random selection scheme over 500 independent trials. The number of CUEs and VUEs were 30 and 60, respectively. Figure 5 studies the cumulative distribution function (CDF) of the total CUEs' capacities. In Figure 5, our proposed algorithms with $\alpha = 0$, $\alpha = 0.5$, and $\alpha = 1$ all obtained better performance in the capacity performance, which reflected the excellent optimality-finding

capability of our proposed algorithm. Moreover, as $\alpha$ decreased, the maximum achievable total CUEs' capacities of our proposed algorithm increased while the corresponding minimum capacity value decreased, illustrating that the decrease of $\alpha$ deteriorated the allocation fairness of the system.

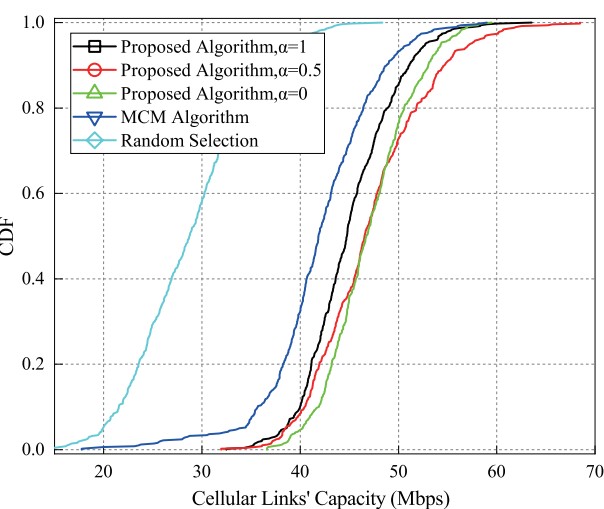

**Figure 5.** The CDF of achievable total CUEs' capacities.

*5.3. Impact of Number of V2I Links*

In this simulation, a vehicular network with different V2I links was considered to evaluate the fairness, transmission capacity, and links' outage performance of our proposed algorithm. The number of VUEs was set to 60, and the number of V2I links ranged from 30 to 100. The simulation results were obtained by 200 independent trials.

As observed in Figure 6, increasing the number of V2I links brought a fairness decline. This was because the VUEs and CUEs could only transmit their messages through the sharing spectrum. As the number of V2I links increased, a portion of CUEs were not allocated to the RBs, leading to the degradation of the system's fairness. For our proposed algorithm, as $\alpha$ increased, the fairness also increased. The fairness gaps between $\alpha = 0$ and $\alpha = 0.5$ were negligible due to the comparable computation efficiency in the preference metric (14) for these two situations. However, when $\alpha$ was equal to zero, the fairness performance of our proposed algorithm declined significantly. This was due to the fact that the $\alpha$-fair function was out of effectiveness when $\alpha = 0$, and in each matching process, our proposed algorithm with $\alpha = 0$ allocated as many RBs as possible to the users with good channel conditions. In the random selection scheme, each VUE randomly selected the CUEs with equal probability. Thus, the fairness performance of the random selection scheme outperformed our proposed algorithm with $\alpha = 0$. Since the MCM algorithm maximized the minimum network capacity through the iterative Hungarian algorithm and the Hungarian algorithm still searched the allocation results maximizing the network capacity, the fairness performance of the MCM algorithm was worse than that of our proposed algorithms with $\alpha = 0.5$ and $\alpha = 1$.

In Figure 7, we compare the achievable total CUEs' capacities of the five algorithms under different V2I link numbers. As the number of V2I links increased, the capacity performance of our proposed three algorithms with different $\alpha$ values increased. This was because increasing the number of V2I links provided more choices for VUEs in their matching process. VUEs could match the CUEs with better channel conditions, improving the total CUEs' transmission capacities. In contrast to the results in Figure 6, our proposed algorithms' achievable total capacity performance decreased as the $\alpha$ increased. Under the small $\alpha$ condition, the $\alpha$-fair function (10) at the same ergodic capacity possessed a larger slope value. Hence, VUEs with large transmission capacities were more favored in the matching process.

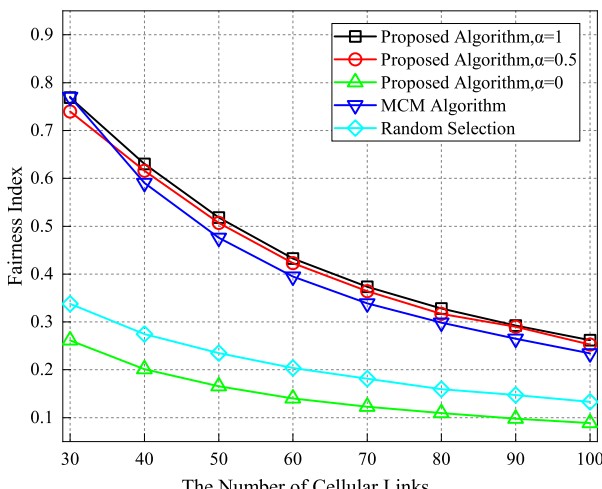

**Figure 6.** The fairness performance of different schemes with varying V2I links.

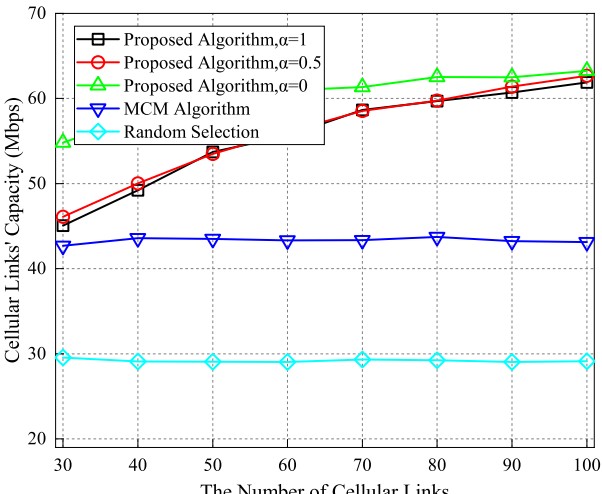

**Figure 7.** The total capacity performance of different schemes with varying V2I links.

Moreover, it is noted that the total transmission capacity performance of the MCM algorithm and the random selection schemes were slightly affected by the number of V2I links. Due to the iterative process of the Hungarian algorithm in the MCM algorithm, the final allocation result was limited by the VUE with the worst channel conditions in the network system regardless of the V2I links' numbers. In the random selection scheme, each VUE only randomly matched one CUE. Therefore, the total transmission capacity performance of the random selection scheme was bounded by the number of VUEs.

Figure 8 demonstrates the vehicular link's outage ratio performance of the five schemes under different numbers of V2I links. As shown in (8) and (9), the LVP performance was positively related to the SINR of the V2V link. Thus, in our proposed algorithms, with the increase in the number of V2I links, the average SINR of the V2V links also increased, improving the links outage ratio performance. Furthermore, the smaller the value of $\alpha$, the better the outage ratio performance. This was because the reduction of the $\alpha$ value increased the proportion of the link's capacity in the VUEs' preference profiles; hence, the average SINR of V2V links rose. In accordance with the analysis in Figure 7, the vehicular link's outage ratio performances of the MCM algorithm and the random selection schemes were less affected by the number of V2I links.

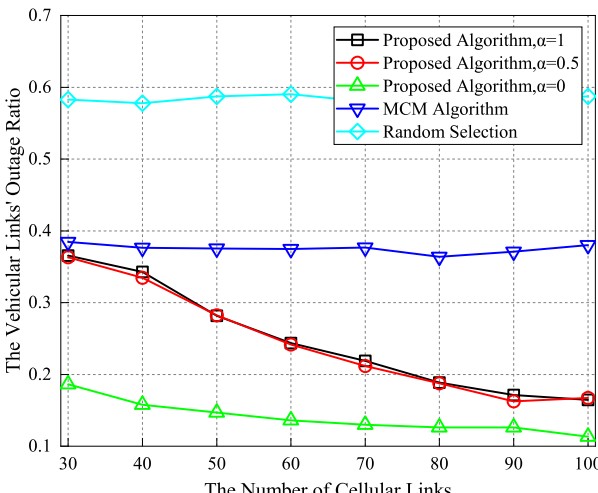

**Figure 8.** The vehicular link's outage ratio performance of different schemes with varying V2I links.

### 5.4. Impact of Number of V2V Links

In this subsection, we evaluated the fairness, transmission capacity, and the link's outage ratio performance of our proposed algorithm under the dynamic V2V links. The number of V2I links was 30, and the range of VUEs was [30, 100]. The results were obtained by simulating 200 independent trials.

Figure 9 studies the fairness performance of the five schemes under dynamic V2V links and shows that there was a marked rise in the fairness performance for all schemes with the increase of V2V links. For our proposed algorithms with $\alpha = 0.5$ and $\alpha = 1$ and the MCM algorithm, their fairness performance rapidly increased when the number of VUEs was less than 60. When the number of VUEs ranged from 30 to 60, the number of VUEs was less than that of V2I links. Thus, in this stage, more CUEs could be assigned to the RBs by reusing the spectrum resources with the V2V links as the number of VUEs increased, leading to a rise in fairness performance. When the number of VUEs was larger than 60, the fairness performance of our proposed algorithm with $\alpha = 1$ and the MCM algorithm could still rise. However, the reasons behind this phenomenon for these two algorithms were different. Since the exchanged preference profiles and the powerful fairness-guaranteed capability when $\alpha = 1$, our proposed algorithm with $\alpha = 1$ could allocate the RBs to the V2I link with less transmission capacity, achieving excellent fairness, while the MCM algorithm only considered the resource allocation of VUEs with the same number of V2I links. Although this allocation method could ensure fairness performance, it was not conducive to improving the total system transmission capacity. For our proposed algorithm with $\alpha = 0$ and the random selection, increasing the number of VUEs provided more reused modes for the CUEs, which could improve the fairness performance.

Figure 10 depicts the impact of the number of VUEs on the capacity performance of the five allocation schemes. As can be observed in Figure 10, the CUEs' total transmission capacities for all algorithms except the MCM algorithm increased linearly with the number of V2V links since an increase in V2V links could accordingly provide more RBs for the V2I links. In addition, as $\alpha$ increased, the optimality finding capability of our proposed algorithm was enhanced. For the MCM algorithm, when the number of V2V links was larger than that of V2I links, the total available RBs in the system remained unchanged since the algorithm in [8] limited the number of V2V links and V2I links to be equal. Moreover, to pursue a higher fairness, the Hungarian algorithm was iteratively executed in the algorithm of [8], leading to overly stable allocation results. Thus, when the number of VUEs reached 60, the capacity performance was no longer affected by the number of V2V links.

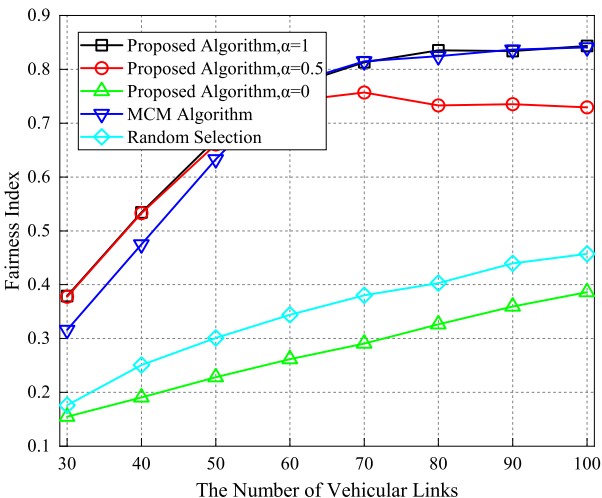

**Figure 9.** The fairness performances of different schemes with varying V2V links.

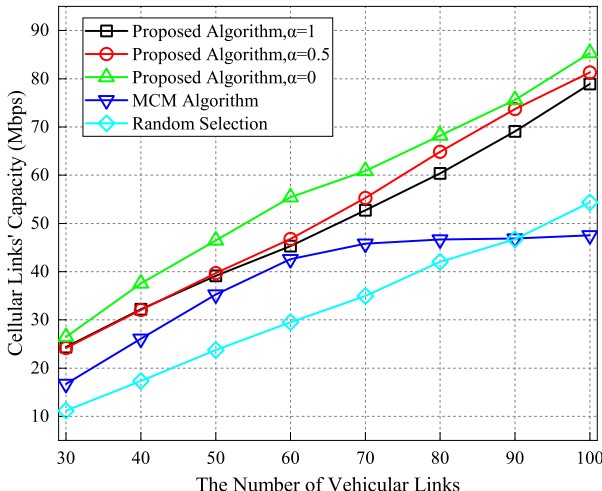

**Figure 10.** The total capacity performance of different schemes with varying V2V links.

In Figure 11, we compare the V2V links' outage ratio of the five schemes under different V2V link numbers. Except for the random selection, increasing the number of V2V links degraded the outage ratio performance of the allocation schemes. This was because the number of V2V links with bad channel conditions increased when the set of VUEs enlarged. In addition, our proposed algorithm with three $\alpha$ values had better performance in outage ratio performance compared to the MCM algorithm and the random selection. When the number of VUEs reached 60, the outage ratio performance of the algorithm in [8] declined sharply since the MCM algorithm did not consider the resource allocation problem for the extra VUEs. In this paper, we also regarded the extra V2V links as the outage links. Due to the randomness in the resource allocation process, the outage ratio of the random selection scheme was maintained at a poor level and was less affected by the number of V2V links.

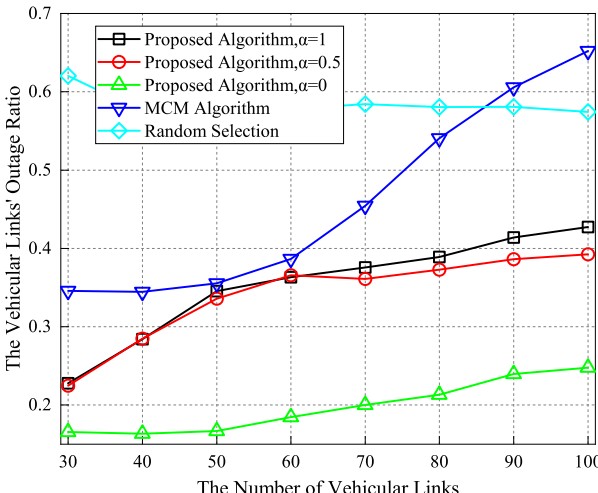

**Figure 11.** The vehicular link's outage ratio performance of different schemes with varying V2V links.

## 6. Conclusions

In this paper, we investigated the RB allocation problem in the C-V2X mode 3 considering the different QoS requirements of VUEs and CUEs. We first established the system communication model and introduced the effective capacity and queuing theory to characterize the V2V link's maximum constant service data rate under the constraint of latency requirement. Then, we formulated the joint power control and RB allocation problem based on the $\alpha$-fair function to maximize the CUEs' total capacities with an $\alpha$-fair function, while guaranteeing the allocation fairness and the reliability of each V2V link. To solve this formulated problem, we proposed a novel matching-game-theoretic algorithm based on the exchanged preference profiles of the two participant sets. In the simulation, we investigated the effects of the $\alpha$ parameter and different structures of the vehicular network on the CUEs' capacities, fairness among CUEs, and V2V links' outage ratio. Simulation results demonstrated that our proposed algorithm was greatly affected by the $\alpha$ parameter and outperformed other resource allocation algorithms.

**Author Contributions:** Conceptualization, D.Y. and D.H.; methodology, D.Y.; software, D.Y.; validation, D.Y.and D.H.; formal analysis, D.Y.; investigation, D.Y., D.H. and X.C.; resources, X.C.; data curation, X.C.; writing—original draft preparation, D.Y.; writing—review and editing, D.Y., D.H. and X.C.; visualization, D.Y.; supervision, X.C.; project administration, X.C. All authors have read and agreed to the published version of the manuscript.

**Funding:** This research received no external funding.

**Conflicts of Interest:** The authors declare no conflict of interest.

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
