# Peer review of "Resource Allocation in C-V2X Mode 3 Based on the Exchanged Preference Profiles"

_electronics, doi:10.3390/electronics12051071_

Round 1
Reviewer 1 Report
Allocation of resource blocks in case of cellular-vehicle-to-everything scenario is addressed.
A game theory algorithm based on the exchanged preference profiles between the two participant sets is proposed.
Results show better efficiency and fairness trade-offscompared to the classic allocation method.
Quality of Service is measured in case of cellular equipment. Queing thoery is employed.
They used alpha fair function. The choice alpha is the key to the success of the method.
Author Response
Thank you for your previous comments! We have adapted the structure of Section 1 and carefully proofread the manuscript. The editorial errors have been corrected and the simulation scenario setup has been supplied, which is number Fig. 4. We have updated the reference and cited the relevant references from MDPI.
Reviewer 2 Report
Good work and nice contribution. Very well written except for minor English issues that can be improved to make the reader engaged.
- Revise the English on some sentences e.g., "Simulation results show that our proposed algorithm is dynamic vehicular network adaptive and better efficiency and fairness trade-offs achieved, outperforming the classic allocation method". Also, change the words "service ability" to one word "serviceability".
- Make sure all acronyms are explained e.g., (SINR) or what does it stand for if it is a well-known acronym?
- Some paragraphs' lines are not numbered. make sure they are done right if using latex and that they are not in the caption section. e.g.,(As can be observed in formula (2), the SINR of the V2V link is dynamic in different slots, leading to the instability of the network...etc.
- It would be very helpful to insert a figure to depict the simulation scenario with BS and Cars with CUE and VUE, on an 800m segment.

Reviewer 3 Report
This paper is interesting and well-constructed in general. This topic is appropriate to the profile of Electronics. I have no remarks on the conducted research including their presentation. But I feel that the literature background and its connection with the study need improvements.
Firstly, the literature list should be slightly refreshed (there are only two positions from the year 2022). Second, and more importantly, research sources should come from different journals and publishing groups. Especially, the citation from “Electronics” and other journals from MDPI Group will enhance the background of the research and will better connect with the topics of the journal to which this manuscript is applied. For example, searching with the use of the keywords “resource allocation fairness” indicate 8 publications from “Electronics” (including two from the year 2022). The keywords “matching game theory” indicates 23 papers in the whole MDPI Group journals set (one from the year 2023). I request for considering some from them.
These aspects should be added in section 1, probably with the division of this section into two: “Introduction” and “Literature review”, and in the new section “Discussion” added before “Conclusions”. The section “Discussion” should contain a broader comparison of the conducted research and their results with the former research cited in the “Literature review” part (including the newly added positions). This section should also highlight the novelty of the research, its limitations, and future works.
Some smaller remark: please, add a figure presenting the highway segment considered in section 5.1 (page 10) with the parameters possible to show.
Author Response
Please see the attachment。

Round 2
Reviewer 3 Report
I am reporting that all of my remarks were effectively considered and I am satisfied. I have no additional remarks and I recommend this paper to publish in “Electronics”.